# ICMOS: INCREMENTAL CONCEPT MINING FOR OS KERNEL CONFIGURATION VIA LLMS AGENTIC REASONING

## ABSTRACT

Linux kernel configuration is critical for system performance, security, and adaptability. However, its vast configuration space, comprising over 17,000 configuration options, renders manual tuning both time-consuming and prone to errors. Existing methods largely rely on static heuristics or limited semantic rules, which struggle to capture complex configuration dependencies or adapt across diverse workloads. We introduce **ICMOS**, a framework that integrates large language models (LLMs) with a heterogeneous knowledge graph of kernel configuration concepts (**OSKC-KG**). By grounding LLM reasoning in structured semantics, ICMOS supports context-aware mining of configuration concepts and agentic concept evolution in response to new requirements and kernel updates. We evaluate ICMOS on configuration QA tasks and diverse real-world workloads, including databases, web servers, in-memory caches, and system benchmarks. ICMOS consistently outperforms LLM-only baselines, delivering higher accuracy, faster optimization, and robust system performance. Notably, it halves optimization time, reduces tail latency by 58.1%, and more than doubles configuration success rates. These results demonstrate that ICMOS provides a scalable and reliable framework for grounding LLM reasoning in structured semantics, thereby advancing kernel configuration understanding and optimization.

## 1 INTRODUCTION

Operating System Kernel Configuration (OSKC) refers to the process of selecting, understanding, and optimizing the configuration items that determine the kernel's behavior and structure. This process is critical to achieving system performance, security, and compatibility (Figure 1). However, modern Linux kernels (Torvalds & Contributors, 2025) comprise over 17,000 options with intricate dependencies, forming an intractably large configuration space that renders manual tuning infeasible. This complexity has spurred various automated solutions; however, existing approaches, such as Kmax (Gazzillo, 2019), rely mainly on static heuristics or manual analysis, fail to capture semantic dependencies, and lack generalization across diverse hardware and workloads.

Although recent work has explored the use of large language models (LLMs), such as GPT-4 (OpenAI et al., 2023), and LLM-based frameworks like AutoOS (Chen et al., 2024) for kernel configuration optimization, several key challenges remain. First, configuration options often lack well-defined semantics, making it difficult for both human experts and LLMs to accurately infer their system-level implications. Second, LLMs are prone to hallucination (Xu et al., 2024), often producing invalid or suboptimal suggestions due to inadequate knowledge of the specific domain. Third, both rule-based and LLM-driven approaches exhibit significant inefficiencies. For instance, AutoOS demands 1 to 2 hours per optimization iteration and often fails to compile due to configuration conflicts.

To address these challenges, we propose **ICMOS**, a novel framework that synergizes LLMs with a heterogeneous knowledge graph for incremental concept mining in kernel configuration spaces. By combining the linguistic comprehension capabilities of LLMs with knowledge graphs' structured constraint modeling, ICMOS achieves configuration-aware reasoning, automatic semantic concept mining, and continuous evolution of mappings between configurations and concepts. Specifically, the foundation of ICMOS is **OSKC-KG**, a heterogeneous graph representation that systematically

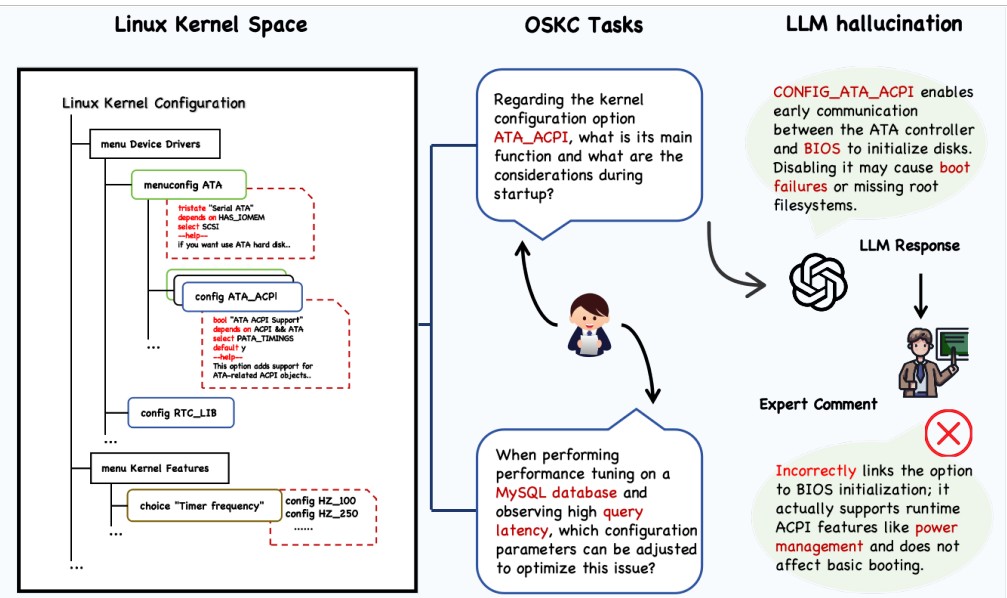

Figure 1: Overview of OSKC: (Left) Linux kernel space and configurations, (Middle) user-centric OSKC tasks, (Right) LLM hallucination challenges.

models kernel configuration spaces through: a *Configuration Graph* that models the hierarchical and dependency structures of kernel configuration items and a *Concept Taxonomy Graph* that captures the evolving functional semantics of configurations. Building upon this representation, ICMOS operates through a two-phase process: (1) **LLM-KG Synergy Mining**, Where the LLM traverses the configuration graph via structured context reasoning to mine functional semantics of configuration items and establish mappings to the concept taxonomy graph; (2) **Agentic Concept Evolution**, where the LLM agent dynamically incorporates user requirements or emerging kernel functions through concept-oriented agentic reasoning. Furthermore, our framework supports human-in-the-loop verification to ensure the correctness, interpretability, and functional relevance of mined concepts.

To validate ICMOS's ability to ground LLM reasoning in structured semantics, we further introduce a suite of downstream *OSKC Understanding Tasks*, including retrieval QA and performance tuning across both *system-level* metrics (e.g., CPU computation, file I/O) and *application-level* workloads (e.g., databases, web servers, in-memory caches). We evaluate ICMOS on these tasks and find that it consistently outperforms LLM-only baselines, achieving substantial gains in accuracy, system performance, and optimization efficiency. Notably, it reduces optimization time by nearly half, more than doubles the configuration success rate, and cuts tail latency by up to 58.1% across representative workloads, underscoring its practicality and reliability.

To summarize our contributions:

1. We present **ICMOS**, the novel framework that integrates large language models with a heterogeneous knowledge graph (**OSKC-KG**) for kernel configuration. This enables structured, configuration-aware reasoning beyond heuristic or LLM-only approaches.

2. We introduce **incremental concept mining**, a novel approach that systematically extracts, organizes, and evolves the semantic concepts of kernel configuration options, establishing a dynamic and extensible concept taxonomy of OSKC semantics.

3. We design a two-phase process that combines *LLM-KG synergy mining* for context-aware semantic mapping and *agentic concept evolution* for adapting to user requirements and emerging kernel features, with human-in-the-loop verification for correctness and interpretability.

4. We establish a benchmark of *OSKC Understanding Tasks* and empirically demonstrate that ICMOS delivers substantial improvements in efficiency and performance over LLM-only baselines across real-world scenarios.

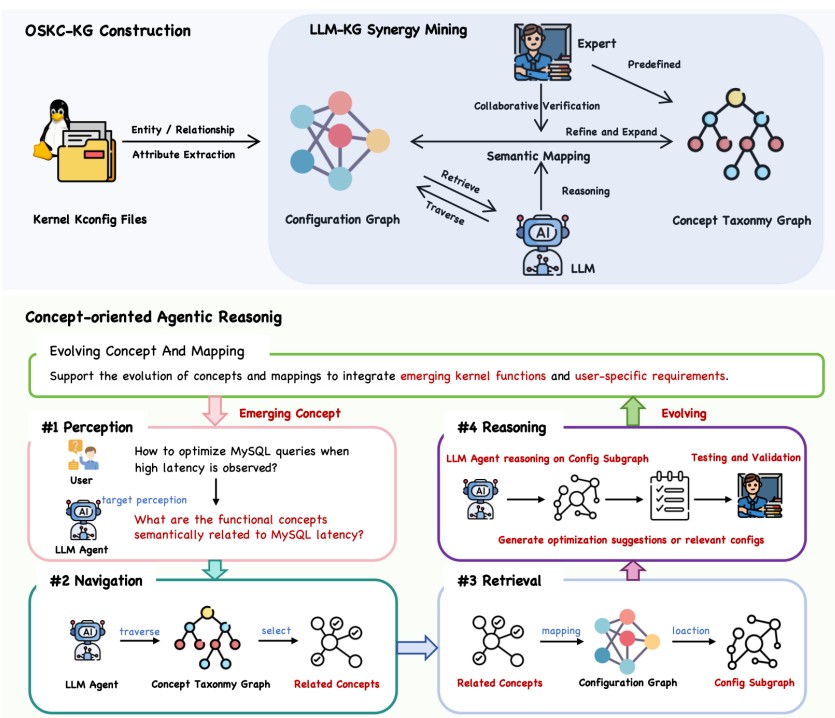

Figure 2: Overview of the ICMOS framework, which integrates OSKC-KG, LLM-KG synergy mining, and agentic concept evolution with human-in-the-loop verification.

## 2 ICMOS FRAMEWORK

We formalize the OSKC challenge as structuring configuration items, associating them with functional semantics, and enabling generalization to downstream tasks such as semantic QA and performance tuning. To address this, we propose **ICMOS**, an incremental concept mining framework that bridges raw kernel configuration options with high-level functional semantics through three synergistic components: (i) OSKC-KG, a heterogeneous knowledge graph unifying configuration structures and functional semantics; (ii) LLM-KG synergy concept mining, for automated, constraint-aware alignment of configuration options to semantic concepts; and (iii) Agentic concept evolution, enabling dynamic adaptation of semantics via concept-driven reasoning. The entire pipeline is reinforced by human-in-the-loop verification to ensure semantic correctness and interpretability (see Figure 2).

### 2.1 OSKC CONCEPT KNOWLEDGE GRAPH

To bridge raw kernel configuration options with high-level functional semantics, we construct the *OSKC-KG*, a heterogeneous knowledge graph formally defined as $\mathcal{G}_{OSKC} = (\mathcal{V}, \mathcal{R}, \mathcal{T})$. It integrates two subgraphs:

**Configuration Graph.** The configuration graph captures structural and logical dependencies among Linux kernel options. Its nodes represent represent configuration entities (e.g., `menu`, `config`, `menuconfig`, `choice`), while its edges encode hierarchical structure (`SUB_OPTION`), dependency constraints (`DEPENDS`), and enforced implications (`SELECTS`). We automatically construct this graph by parsing the official Linux kernel's `Kconfig` files using `Kconfiglib` (Rouberg et al., 2018), yielding a scalable representation of hierarchical and dependency structures across kernel versions. Details of parsing, node extraction, and relation definitions are provided in the Appendix B.1.1.

**Concept Taxonomy Graph.** Modern kernel configuration options are often encoded in cryptic or hardware-specific identifiers, making semantic interpretation difficult for both human developers and LLMs. For example, the option `ATA_ACPI` appears obscure in isolation, yet its source description

reveals associations with both *Power Management* and *Storage Support*. Such latent semantics are essential for managing and optimizing large-scale configuration profiles. To address this challenge, we construct a hierarchical *Concept Taxonomy Graph* that systematically organizes kernel configurations' functional semantics. The taxonomy is seeded with expert-defined coarse-grained domains (e.g., *Security Features*, *Performance*, *Hardware Support*) and dynamically expanded via LLM-guided fine-grained concept mining (see following paragraph). This hybrid construction ensures both semantic coverage and extensibility, providing an interpretable space for aligning raw configuration items with high-level abstractions. An illustrative subset is shown in Table 1, while formal definitions and the complete construction process are included in Appendix B.1.2.

**Unified Representation.**    Crucially, OSKC-KG is not a simple juxtaposition of the two subgraphs. Instead, LLM-guided concept mining leverages structured information from the configuration graph to infer functional semantics, dynamically expanding the taxonomy and establishing cross-subgraph mapping relations. These mappings unify the two subgraphs into a coherent heterogeneous knowledge graph that supports interpretable reasoning and downstream tasks.

Table 1: Part of Hierarchical Concept Taxonomy

| Expert-defined Coarse Concepts | | LLM-mined |
| --- | --- | --- |
| **Top-level** | **Sub-level** | **Fine-grained Concepts** |
| Core Subsystem | Filesystem, Networking | Flash Filesystem, Wireless |
| Security Features | Access Control, Cryptography | Process Signal Control |
| Hardware Support | Storage, Peripheral | SCSI, Bus Support |
| Performance | Memory, I/O Optimization | PMD, DMA Optimization |

**LLM-KG Synergy Concept Mining.**    To bridge the configuration graph with the concept taxonomy graph and incrementally mine fine-grained functional semantics, we propose an LLM-KG synergy mining process. The core mechanism leverages a traversal of the configuration graph to extract rich contextual signals, including structural dependencies and natural language descriptions, and enables the LLM to predict appropriate semantic concepts. This process establishes interpretable mappings between configuration items and concepts while dynamically enriching the concept taxonomy.

We adopt a *Hierarchical Hybrid Traversal (HHT)* strategy to provide structure-aware context for the LLM. First, a global breadth-first search (BFS) identifies branching nodes (menu, menuconfig, choice) and assigns depth levels. Second, local depth-first search (DFS) collects descendant config nodes into *branch units*, enriched with two key contexts: (1) a *parent state* encoding high-level intent from ancestors, and (2) a *sibling state* aggregating peer semantics for contrastive reasoning. This structured context enables the LLM to produce coherent, taxonomy-aligned concept mappings.

The semantic mapping set is defined as:

$$\mathcal{M} \subseteq \mathcal{C}_{ext} \times \mathcal{V}_{cfg}, \tag{1}$$

where $\mathcal{C}_{ext}$ is the extended concept taxonomy (expert-defined and LLM-mined), and $\mathcal{V}_{cfg}$ is the set of configuration nodes. Each mapping is further annotated with a confidence score, rationale, and hierarchical insertion rule (see Appendix B.2 for details). For each configuration node $v \in \mathcal{V}_{cfg}$, we retrieve structured context from the configuration graph to guide the semantic alignment:

$$\mathcal{K}_v = \{\mathcal{P}_v, \mathcal{D}_v, P_\ell, S_\ell\}, \tag{2}$$

comprising: (1) the hierarchical path $\mathcal{P}_v$, (2) the configuration node description $\mathcal{D}_v$, (3) the parent state $P_\ell$, and (4) the sibling state $S_\ell$. This context enables the LLM to perform grounded, consistent reasoning by leveraging structural dependencies, mitigating hallucination and ambiguity.

Given the context $\mathcal{K}_v$, the LLM predicts a set of relevant concepts from $\mathcal{C}_{ext}$. The top predictions, filtered by a confidence threshold $\theta$, are used to establish RELATED_TO mappings between the configuration item and its functional concepts. Simultaneously, any newly discovered concepts are integrated into the taxonomy via SUB_CATEGORY relations. All outputs support optional human-in-the-loop validation to ensure semantic accuracy and practical utility. This closed-loop process enables the OSKC-KG to evolve coherently, aligning low-level configurations with high-level semantics in a structured and interpretable manner. Detailed prompt templates and reasoning traces, along with illustrative mined concept examples, are provided in Appendix B.3 and Appendix B.4.

## 2.2 CONCEPT-ORIENTED AGENTIC REASONING

While the synergy mining process aligns configuration items with fine-grained concepts by leveraging the structured semantics of OSKC-KG, real-world requirements and kernel evolution continuously introduce new functionalities, such as workload-specific goals (e.g., MySQL latency optimization, secure container isolation) and kernel features (e.g., advanced scheduling or eBPF integration). To preserve semantic completeness and adaptability, ICMOS enables the incremental integration of emerging concepts into OSKC-KG.

We propose *Concept-oriented Agentic Reasoning*, a framework that leverages LLM agents to extend both the taxonomy $\mathcal{C}_{ext}$ and the semantic mappings $\mathcal{M}$. The process exploits three components of OSKC-KG: the configuration graph, the concept taxonomy, and their mappings, within a structured agentic pipeline.

**Agentic Reasoning Process.** Given a new concept $C_{\text{new}}$, the agent executes a structured four-stage pipeline: *perception, navigation, retrieval, and reasoning* (as illustrated in Figure 2). First, the agent reformulates $C_{\text{new}}$ into a query prompt and performs guided traversal over $\mathcal{C}_{ext}$ to identify semantically related branches. The result is a candidate set of related concepts $\mathcal{T}_{\text{related}}$. Using the existing mapping $\mathcal{M}$, the agent retrieves the associated configurations:

$$\mathcal{V}_{\text{cand}} = \bigcup_{T_j \in \mathcal{T}_{\text{related}}} \{v \,|\, (v, T_j) \in \mathcal{M}\}, \tag{3}$$

which induces a subgraph $\mathcal{G}_{\text{config}} \subseteq \mathcal{V}_{cfg}$ representing the relevant configuration space.

The agent then performs retrieval-augmented reasoning over $\mathcal{G}_{\text{config}}$, leveraging metadata such as help texts, dependency constraints, and selection rules. This produces an increment of semantic mappings:

$$\mathcal{M}_{\text{new}} = \{(v, C_{\text{new}}, s, r) \mid v \in \mathcal{V}_{\text{cand}}\}, \tag{4}$$

where $s \in [0, 1]$ is a confidence score and $r$ denotes the rationale. Finally, the integration of $C_{\text{new}}$ into the taxonomy is guided by semantic alignment and subject to human-in-the-loop validation. If $C_{\text{new}}$ is belongs to an existing branch, it is inserted via a SUB_CATEGORY relation; otherwise, it is added as a new root-level concept. The mapping set is then updated as $\mathcal{M} \leftarrow \mathcal{M} \cup \mathcal{M}_{\text{new}}$.

## 2.3 HUMAN-IN-THE-LOOP VERIFICATION

To ensure the reliability of the semantic mapping $\mathcal{M}$ and the evolving concept taxonomy $\mathcal{C}_{\text{ext}}$, we incorporate a human-in-the-loop verification mechanism that enables domain experts to examine the generated concept assignments by inspecting the associated confidence scores and reasoning traces, and subsequently validate their semantic correctness.

This verification process supports both global refinement of the taxonomy (e.g., merging overlapping concepts or adjusting hierarchical structures) and local validation of individual configuration-to-concept mappings. Furthermore, the refined concepts and mappings undergo *downstream task-driven validation* (Section 3), where experts assess their semantic validity and effectiveness in real-world scenarios, such as configuration optimization and retrieval QA.

## 3 OSKC UNDERSTANDING TASKS

To evaluate the effectiveness of ICMOS and the semantic quality of OSKC-KG, we introduce a suite of downstream *OSKC Understanding Tasks*. These tasks evaluate the interpretability and operational utility of our framework within realistic system deployment contexts.

Formally, given a natural-language query $q$ (e.g., a configuration question or optimization goal), the task is to return either (i) a semantic interpretation $\mathcal{A}_q$, or (ii) a relevant configuration set $\mathcal{K}_q \subseteq \mathcal{K}$. We consider two representative settings:

- **Configuration Retrieval QA**: Given a question (e.g., "Which options relate to process scheduling?"), retrieve the relevant configuration items *and* their associated semantic explanations from OSKC-KG.

- **Intelligent Configuration Optimization**: Given a performance target, generate an optimized configuration set that improves system-level or application-level performance, while respecting kernel dependency constraints.

Together, these tasks form a benchmark for kernel configuration understanding, covering both symbolic reasoning and performance-driven optimization. We validate results through human-in-the-loop evaluation and quantitative comparison against baselines.

## 4 EXPERIMENT

We validate ICMOS on the OSKC Understanding Tasks (Section 3), benchmarking against LLM-only and heuristic baselines. We first present the experimental setup and OSKC-KG characterization, followed by evaluations on *Configuration Retrieval QA* and *Intelligent Configuration Optimization*.

### 4.1 EXPERIMENTAL SETUP AND METRICS

**Setup.** We construct OSKC-KG from the Linux 6.x kernel series (versions 6.1–6.15) and implement ICMOS with *DeepSeek-V3-0324* (DeepSeek-AI et al., 2024) as the core LLM. All data are stored in *Neo4j* (Neo4j Team, 2025) to support efficient traversal, retrieval, and expansion. Optimization experiments are conducted on a virtual machine running Ubuntu 24.04 (Linux 6.8, aarch64) with 4 CPU cores and 4 GB RAM.

**Metrics.** We adopt task-specific metrics: (i) *Configuration Retrieval QA*: tag-level **Recall**, **F1**, **Top-3 Accuracy**, and option-level **Exact Match Rate**. (ii) *Intelligent Configuration Optimization*: **UnixBench** scores (Lucas, 2015) (CPU, memory, I/O, scheduling) for system-level evaluation, and application-level **throughput** and **latency** (e.g., Apache, MySQL, Redis), measured with ApacheBench (Apache Software Foundation, 2013), sysbench (Kopytov, 2023), and memtier_benchmark (Redis Labs, 2024). We further report **tuning efficiency** (optimization time per cycle) and **conflict-free success rate**.

### 4.2 OSKC-KG CHARACTERIZATION

The constructed OSKC-KG comprises 18,927 nodes and 145,191 edges (Table 2), covering 17,624 configuration items and 1,303 semantic concepts. Notably, 96.5% of configurations are semantically annotated, with each item associated with an average of 3.9 concepts (including inherited ones), reflecting high semantic density. This broad coverage, semantic richness, and extensibility, establish OSKC-KG as a scalable foundation for downstream reasoning, retrieval, and optimization. The end-to-end construction of OSKC-KG required approximately 25 hours (graph construction: 1h; concept mining: 16h; expert validation: 8h). These costs are dominated by the initial build; subsequent updates (e.g., new kernel versions or user requirements) are incremental and incur minimal overhead for extraction and verification. Further statistics and the schema overview are provided in Appendix B.5.

Table 2: OSKC-KG Scale and Coverage Statistics

| Type | Count | % |
|---|---|---|
| **Nodes (Total: 18,927)** | | |
| Config / Menuconfig | 17,242 | 91.3 |
| Concept Nodes | 1,303 | 6.8 |
| Menu / Choice Nodes | 382 | 1.9 |
| **Edges (Total: 145,191)** | | |
| DEPENDS | 47,244 | 32.6 |
| SELECTS | 13,919 | 9.6 |
| SUB_OPTION | 17,455 | 12.0 |
| SUB_CATEGORY | 1,370 | 0.9 |
| RELATED_TO | 65,203 | 44.9 |
| **Mapping** | | |
| Mapped Configs/Menuconfigs | 16,630 / 17,242 | 96.5 |
| Avg. Tags per Config | 3.9 | – |

### 4.3 CONFIGURATION RETRIEVAL QA

To evaluate whether OSKC-KG enhances semantic retrieval for kernel configurations, we construct a curated evaluation set of 100 expert-annotated QA pairs, covering two key tasks: (i) *Concept Understanding* and (ii) *Configuration Selection* (see Appendix C for details). As shown in Table 3, OSKC-KG augmentation substantially outperforms the LLM-only baseline: concept recall increases from 66.7% to 88.9%, concept F1 improves by 16.2 percentage points (from 49.2% to 65.4%),

and exact match accuracy for configuration selection rises from 58.7% to 86.9%. These gains demonstrate that structured semantic grounding not only improves retrieval fidelity but also enhances interpretability by aligning model outputs with domain concepts.

Table 3: Performance on Kernel Configuration QA with and without OSKC-KG Augmentation

| Metric | LLM-only | LLM + OSKC-KG |
|---|---|---|
| *Concept Understanding* | | |
| Tag Recall | 66.7% | **88.9%** |
| Tag F1 | 49.2% | **65.4%** |
| Top-3 Accuracy | 46.3% | **75.9%** |
| *Configuration Selection* | | |
| Exact Match Rate | 58.7% | **86.9%** |
| Span-level F1 | 85.9% | **96.0%** |

## 4.4 INTELLIGENT CONFIGURATION OPTIMIZATION

We evaluate ICMOS in practical, deployment-oriented settings, examining both performance gains and reliability under kernel constraints. To ensure that the produced configurations are feasible and deployable, we augment LLM reasoning with a *Conflict-Aware Validation Mechanism* (detailed in Appendix D) that enforces dependency, selection, default-value, and visibility constraints derived from OSKC-KG, thereby preventing infeasible or mutually conflicting option sets.

We assess optimization along two complementary dimensions: **(i) System-level performance**, measuring CPU, memory, I/O, and scheduling efficiency using UnixBench; and **(ii) Application-specific tuning**, evaluating end-to-end performance for representative workloads (web servers, databases, in-memory caches) under high-concurrency benchmarks. This separation enables us to capture both micro-level efficiency and macro-level workload adaptability of the resulting configurations.

**Comparison Setup.** We compare three strategies: (i) *Default*: the official Linux configuration; (ii) *AutoOS*: LLM-only optimization without external knowledge; (iii) *ICMOS (ours)*: the full concept-oriented agentic reasoning pipeline (Section 2.2), augmented with OSKC-KG retrieval and a conflict-aware validation mechanism to ensure semantic grounding and feasibility. All evaluations follow the metrics defined in Section 4.1.

Table 4: UnixBench Results: Baseline vs. AutoOS (LLM-only) vs. ICMOS. (Changes are relative to Baseline. Higher is better for all metrics.)

| Test Case | Baseline | AutoOS | ICMOS |
|---|---|---|---|
| Dhrystone 2 | 8335.2 | 8391.1 (+0.7%) | **8449.8 (+1.4%)** |
| Whetstone | 1586.3 | 1611.7 (+1.6%) | **1626.9 (+2.6%)** |
| Execl Throughput | 2247.9 | 2389.8 (+6.3%) | **2401.0 (+6.8%)** |
| File Copy 1024b | 4873.6 | 5121.4 (+5.1%) | **5177.6 (+6.2%)** |
| File Copy 256b | 3279.1 | 3339.5 (+1.8%) | **3384.3 (+3.2%)** |
| File Copy 4096b | 8164.2 | 9231.4 (+13.1%) | **10152.8 (+24.4%)** |
| Pipe Throughput | 2177.4 | 1963.4 (-9.8%) | **2302.8 (+5.8%)** |
| Context Switch | 78.8 | 86.7 (+10.0%) | 84.4 (+7.1%) |
| Process Creation | 733.0 | 810.1 (+10.5%) | **873.8 (+19.2%)** |
| Shell Scripts (1) | 4826.4 | 5183.9 (+7.4%) | **5266.6 (+9.1%)** |
| Shell Scripts (8) | 10161.7 | 9933.7 (-2.2%) | **10385.2 (+2.2%)** |
| System Call | 1056.2 | 1093.3 (+3.5%) | **1099.7 (+4.1%)** |
| **Index Score** | 2287.6 | 2348.4 (+2.7%) | **2432.1 (+6.3%)** |

**System-Level Performance.** As shown in Table 4, ICMOS consistently improves all twelve UnixBench metrics, achieving +6.3% overall gains versus +2.7% for AutoOS, and up to +24.4% in file I/O and +19.2% in process creation. Unlike AutoOS, which occasionally introduces regressions due to uninformed configuration choices, ICMOS maintains stability by leveraging structural knowledge and conflict validation. A representative set of system-level optimization recommendations generated by ICMOS is shown in Appendix E.

Table 5: Application-Specific Performance: Baseline vs. AutoOS (LLM-only) vs. ICMOS across Web Servers (Apache/Nginx), Databases (MySQL/PostgreSQL), and In-Memory Caches (Redis/Memcached). Evaluated with ApacheBench, sysbench, and memtier_benchmark. (%↑ = improvement over Baseline, Throughput: higher better; Latency: lower better.)

| Metric | Apache | | | Nginx | | |
|---|---|---|---|---|---|---|
| | Baseline | AutoOS (%↑) | ICMOS (%↑) | Baseline | AutoOS (%↑) | ICMOS (%↑) |
| Time Taken for Tests (s) | 0.235 | 0.232,+1.3% | **0.228,+3.0%** | 0.268 | 0.257,+4.1% | **0.253,+5.6%** |
| Requests per Second (req/s) | 42,490 | 43,095,+1.4% | **43,816,+3.1%** | 37,324 | 38,916,+4.3% | **39,449,+5.7%** |
| Time per Request (ms) | 2.354 | 2.320,+1.4% | **2.282,+3.1%** | 2.679 | 2.570,+4.1% | **2.534,+5.4%** |
| Transfer Rate (KB/sec) | 454,155 | 460,614,+1.4% | **468,332,+3.1%** | 397,881 | 413,555,+3.9% | **420,533,+5.7%** |
| Max Request Time (Total, ms) | 13 | 19,-46.2% | 17.5,-34.6% | 12 | 20,-66.7% | **11,+8.3%** |

| Metric | MySQL | | | PostgreSQL | | |
|---|---|---|---|---|---|---|
| | Baseline | AutoOS (%↑) | ICMOS (%↑) | Baseline | AutoOS (%↑) | ICMOS (%↑) |
| Transactions per Second (TPS) | 1,919.5 | 1,952.2,+1.7% | **1,996.9,+4.0%** | 556.2 | 560.4,+0.8%, | **617.1,+11.0%** |
| Queries per Second (QPS) | 39,502 | 40,191,+1.7% | **41,058,+3.9%** | 11,489 | 11,570,0.7% | **12,744,+10.9%** |
| Average Latency (ms) | 52.08 | 51.20,+1.7% | **50.06,+3.9%** | 53.92 | 53.71,+0.4% | **47.96,+11.1%** |
| 95th Percentile Latency (ms) | 111.67 | 112.00,+0.3% | **104.84,+6.1%** | 24.83 | 22.28,+10.3% | **18.28,+26.4%** |

| Metric | Redis | | | Memcached | | |
|---|---|---|---|---|---|---|
| | Baseline | AutoOS (%↑) | ICMOS (%↑) | Baseline | AutoOS (%↑) | ICMOS (%↑) |
| Total Throughput (ops/sec) | 257,552 | 267,080,+3.7% | **270,841,+4.0%** | 447,110 | 461,583,+3.2% | **484,847,+8.4%** |
| Average Latency (ms) | 1.56 | 1.50,+3.7% | **1.48,+4.6%** | 0.90 | 0.88,+2.8% | **0.88,+2.9%** |
| P50 Latency (ms) | 1.46 | 1.40,+4.2% | 1.49,-1.1% | 0.57 | 0.60,-4.7% | **0.52,+8.4%** |
| P99 Latency (ms) | 3.31 | 3.28,-1.0% | **3.07,+7.3%** | 3.97 | 3.73,+5.9% | 3.96,+0.1% |
| P99.9 Latency (ms) | 11.92 | 18.18,-52.4% | **4.99,+58.1%** | 5.92 | 6.82,-5.1% | 6.91,-16.8% |
| Throughput (KB/sec) | 74,833 | 77,601 (-3.7%) | **78,694,+5.2%** | 130,286 | 134,504,+3.2% | **137,616,+5.6%** |

**Application-Specific Tuning.** Across six real-world workloads (Table 5), ICMOS yields substantial benefits, reducing tail latency by up to 58.1% and improving throughput by as much as 11.0%. For example, PostgreSQL achieves 26.4% lower P95 latency alongside notable QPS gains (+10.9%). In contrast, AutoOS often degrades performance (e.g., increased Nginx request time), highlighting the robustness of ICMOS across diverse workloads.

**Efficiency and Reliability.** As summarized in Table 6, ICMOS reduces optimization time by 50% (39 min vs. 82 min per cycle) and increases configuration success rate from 29% to 76%, while nearly eliminating compilation/boot failures. These results confirm that conflict-aware reasoning not only enhances performance but also ensures reliable and deployable configurations.

Table 6: Optimization Efficiency and Reliability: AutoOS vs. ICMOS. Improvements (Imp.%) are computed relative to AutoOS; lower is better for time and conflicts, higher is better for success rate.

| Metric | AutoOS | ICMOS | Imp. (%) |
|---|---|---|---|
| Average Optimization Time per Cycle | 82 minutes | **39 minutes** | **47.6%** |
| Average Conflicts per Iteration | 4.5 | **1.4** | **68.9%** |
| Configuration Success Rate | 29% | **76%** | **162.1%** |

## 4.5 ABLATION STUDY

While Sections 4.3–4.4 conclusively demonstrate that augmenting LLMs with OSKC-KG yields significant gains over the LLM-only AutoOS baseline, we now isolate the contribution of *conceptual completeness* in the taxonomy.

Specifically, we design a **concept masking** experiment: we randomly remove 30% of mid- and leaf-level nodes from the taxonomy, along with their edges and mappings, simulating scenarios with partial semantic coverage. This directly limits the contextual guidance available to the LLM agent during optimization.

As reported in Table 7, the masked variant (ICMOS-Masked) still outperforms the LLM-only baseline, reflecting the robustness of our agentic reasoning process. However, it consistently underperforms the full ICMOS model, particularly in benchmarks requiring fine-grained semantic differentiation.

Table 7: Ablation: Impact of concept masking on UnixBench performance. ICMOS-Masked removes 30% of mid- and leaf-level concepts and their mappings.

| Test Case | Baseline | AutoOS | ICMOS Masked | ICMOS |
|---|---|---|---|---|
| Dhrystone 2 | 8335.2 | 8391.1 | 8444.2 | **8449.8** |
| Whetstone | 1586.3 | 1611.7 | 1607.8 | **1626.9** |
| Execl Throughput | 2247.9 | 2389.8 | 2378.4 | **2401.0** |
| File Copy 1024b | 4873.6 | 5121.4 | 5120.6 | **5177.6** |
| File Copy 256b | 3279.1 | 3339.5 | 3342.7 | **3384.3** |
| File Copy 4096b | 8164.2 | 9231.4 | 8893.0 | **10152.8** |
| Pipe Throughput | 2177.4 | 1963.4 | 2200.8 | **2302.8** |
| Context Switch | 78.8 | 86.7 | 80.5 | **84.4** |
| Process Creation | 733.0 | 810.1 | 755.5 | **873.8** |
| Shell Scripts (1) | 4826.4 | 5183.9 | 4669.6 | **5266.6** |
| Shell Scripts (8) | 10161.7 | 9933.7 | 10268.3 | **10385.2** |
| System Call | 1056.2 | 1093.3 | 1063.7 | **1099.7** |
| **Index Score** | 2287.6 | 2348.4 | 2376.4 | **2432.1** |

These results indicate that a complete concept taxonomy substantially improves contextual reasoning and optimization effectiveness, and that structured semantic knowledge is essential for unlocking the full potential of LLM-guided kernel tuning.

To further illustrate the practical utility of ICMOS, we conduct a case study on workload-specific optimization (details in Appendix F).

## 5 RELATED WORK

**Traditional Kernel Configuration.** Early kernel configuration (OSKC) optimization methods rely on static analyses and heuristics over Kconfig files. Representative tools such as Kmax (Gazzillo, 2019) validate configuration constraints, while learning-based systems like VConf (Rao et al., 2009) and DeepPerf (Ha & Zhang, 2019) employ probabilistic models and deep networks to predict performance or reduce kernel size. More recent transfer learning approaches (Herzog et al., 2021; Martin et al., 2021) improve generalization across hardware but remain data-intensive and lack semantic interpretability. Although Hou et al. (Pengpeng et al., 2021) propose a visual multi-label graph for configuration understanding, this approach depends on manual curation and lacks scalability.

**LLM-based Configuration Optimization.** LLM-driven systems such as AutoOS (Chen et al., 2024) explore kernel tuning via prompt-based reasoning and generation. While effective for initial exploration, these methods often lack deep semantic grounding of configuration options, leading to ad hoc heuristics, hallucination risks, and limited interpretability.

**Structured Knowledge and Agent-based Reasoning.** To improve factuality, retrieval-augmented generation (RAG) (Lewis et al., 2020) and its graph extensions (e.g., GraphRAG (Han et al., 2024)) leverage structured knowledge to guide LLM reasoning. Similarly, agent frameworks such as ReAct (Yao et al., 2022) and SWE-agent (Yang et al., 2024) combine reasoning with tool use for autonomous decision-making. However, existing approaches rarely incorporate structured semantics in system-level domains such as OS kernel configuration. Our work bridges this gap by integrating LLM agents with a domain-specific knowledge graph, enabling semantically grounded, configuration-aware reasoning for kernel optimization.

## 6 CONCLUSION

We presented ICMOS, a novel framework that grounds LLM reasoning in the structured semantics of OSKC-KG to enable scalable, interpretable kernel configuration. By unifying configuration structure with dynamic concept taxonomies, ICMOS supports incremental concept mining and agentic evolution. Experiments demonstrate consistent gains over LLM-only baselines in both retrieval and optimization tasks, improving semantic fidelity, system performance, and reliability. Looking ahead, we will extend ICMOS to broader kernel versions and heterogeneous environments, further advancing knowledge-driven system automation.

## ETHICS STATEMENT

This research complies with the ICLR Code of Ethics. Our study focuses on operating system kernel configuration and optimization using knowledge graphs and large language models. It does not involve human subjects, sensitive personal data, or proprietary/confidential information. All data used for constructing OSKC-KG are derived from publicly available Linux kernel sources (Torvalds & Contributors, 2025), and the experimental environments are fully reproducible. We believe that the proposed methods pose no foreseeable risks of harm, bias, or misuse, and that they contribute positively to advancing system-level optimization and interpretability research.

## REPRODUCIBILITY STATEMENT

We have taken multiple steps to ensure the reproducibility of our work.

- **Code and Framework.** The implementation of ICMOS, including data preprocessing, OSKC-KG construction, and evaluation pipelines, is provided as anonymized supplementary material and will be fully open-sourced upon acceptance.
- **Data.** The expert-annotated QA dataset used in Section 4.3, along with processing scripts, is included in the supplementary material and will also be released publicly after acceptance.
- **Experiments.** Experimental setup (hardware, software, benchmarks) and evaluation metrics are provided in Section 4. Hyperparameters, prompts, and additional results (including ablation and case studies) are reported in Appendix E and Appendix F.
- **Theoretical and Algorithmic Details.** Formal definitions and algorithmic procedures (e.g., semantic mapping, traversal strategy, and agentic reasoning) are presented in Sections 2.1–2.2, with more detailed explanations and extended discussions provided in Appendix B.

Together, these resources provide sufficient detail for independent reproduction of both our methodology and results.

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

## A    LLM Usage Disclosure

In accordance with ICLR 2026 submission policies, we disclose the usage of large language models (LLMs) in the preparation of this paper. LLMs (e.g., ChatGPT) were used solely as a writing aid for language polishing and grammar refinement. They were not involved in research conception, methodology design, implementation, or experimental evaluation. Specifically, LLMs were occasionally used to improve readability in the *Abstract* and *Introduction* sections, without altering the technical content. All ideas, methods, analyses, and results presented in this paper are entirely the work of the authors.

## B    OSKC-KG Construction Details

### B.1    Formal Definition of OSKC-KG

The OSKC-KG is formally defined as a heterogeneous knowledge graph:

$$\mathcal{G}_{\mathrm{oskc}} = (\mathcal{V}, \mathcal{R}, \mathcal{T}) \tag{5}$$

where:

- $\mathcal{V} = V_{\mathrm{cfg}} \cup V_{\mathrm{concept}}$: the union of nodes from the configuration graph and concept taxonomy graph;

- $\mathcal{R} = R_{\mathrm{cfg}} \cup R_{\mathrm{concept}} \cup R_{\mathrm{map}}$: includes intra-subgraph relations (e.g., DEPENDS, SUB_CATEGORY) and inter-subgraph mapping relations (e.g., RELATED_TO);

- $\mathcal{T} \subseteq \mathcal{V} \times \mathcal{R} \times \mathcal{V}$: the set of all valid triples in the graph, where each triple $(h, r, t)$ denotes a semantic or structural relationship from the head entity $h$ to the tail entity $t$ under the type of relation $r$.

This unified representation enables semantic alignment between raw configuration options and high-level functional concepts, facilitating downstream tasks such as configuration recommendation, retrieval QA, and intelligent optimization.

#### B.1.1    Configuration Graph

To systematically model the structural and semantic complexity of Linux kernel configurations, including hierarchical organization, logical dependencies, and inter-option associations, we propose a structured representation called the *Configuration Graph* $\mathcal{G}_{\mathrm{cfg}} = (V_{\mathrm{cfg}}, E_{\mathrm{cfg}})$. The Configuration Graph is formally defined as:

$$\mathcal{G}_{\mathrm{cfg}} = (V_{\mathrm{cfg}}, R_{\mathrm{cfg}}, T_{\mathrm{cfg}}) \tag{6}$$

where:

- $V_{\mathrm{cfg}}$ denotes the set of configuration entities

- $R_{\mathrm{cfg}}$ contains intra-configuration relations (e.g., DEPENDS, SELECTS)

- $T_{\mathrm{cfg}} \subseteq V_{\mathrm{cfg}} \times R_{\mathrm{cfg}} \times V_{\mathrm{cfg}}$ represents valid triples derived from kernel source semantics.

**Corpus Source**    The construction of the Configuration Graph is grounded in the official Linux kernel source, primarily the distributed set of Kconfig files. Each Kconfig file declaratively defines a collection of configuration options, including their types, default values, help descriptions, logical dependencies (e.g., depends on, select), and their hierarchical organization into menu or menuconfig. (Kconfig structure in Figure 1). We use a static analysis tool Kconfiglib as the core parser and build a data processing pipeline on top of it, enabling consistent extraction of configuration items (e.g., config, menu) and their interdependencies across different Linux kernel versions. This pipeline processes Kconfig files to generate structured metadata, which serves as the foundational data for constructing the configuration graph.

**Node Extraction**    Building upon the structured metadata obtained from Kconfig parsing, we systematically identify the configuration entities to construct the node set $V_{\mathrm{cfg}}$. Each node $v \in V_{\mathrm{cfg}}$ represents a distinct configuration element with preserved semantic attributes (e.g., type, default value, help text). We define four primary node types:

- menu: Hierarchical containers grouping related options (e.g., `Device Drivers`).
- config: A basic configurable option (e.g., `CONFIG_RTC_LIB`).
- menuconfig: High-level toggles that expose subordinate options (e.g., `CONFIG_ATA` → `CONFIG_ATA_ACPI`).
- choice: Mutually exclusive groups (e.g., `CONFIG_HZ` timer frequency, where only one option like `CONFIG_HZ_100` can be selected).

These node types are automatically recognized and extracted via our custom-built data pipeline, ensuring scalability across different Linux kernel versions.

**Relation Extraction** Based on the syntactic and semantic patterns in `Kconfig` files, we formally define three core relation types (denoted as $R_{\text{cfg}}$) which model structural and logical dependencies between the configuration nodes:

- SUB_OPTION: Captures parent-child relationships in menu structures (e.g., `menu` or `menuconfig` → `config`).

- DEPENDS: Encodes logical constraints from `depends on` clauses (e.g., `CONFIG_ATA_ACPI depends on CONFIG_ACPI`).

- SELECTS: Enforced implications via `select` clauses (e.g., `CONFIG_ATA_ACPI` forces `CONFIG_PATA_TIMINGS`)

These relationships are automatically extracted during the `Kconfig` parsing phase, preserving the complete constraint semantics from the original configuration system.

The configuration graph provides a structured representation of kernel configuration items and their interdependencies, enabling precise modeling of hierarchical organization, logical constraints, and semantic associations. This modular design further supports dynamic evolution with new configuration nodes across kernel versions.

### B.1.2 CONCEPT TAXONOMY GRAPH

Modern kernel configuration options often encode low-level mechanisms using cryptic or hardware-specific identifiers, which makes semantic interpretation challenging for both human developers and LLMs. For instance, the option `ATA_ACPI` may seem obscure in isolation, but its description (e.g., help text) reveals associations with both *Power Management* and *Storage Support*. These latent semantics are essential to effectively manage, understand, and optimize configuration profiles at scale.

To address this problem, we construct an extensible *Concept Taxonomy Graph* $\mathcal{G}_{\text{concept}} = (V_{\text{concept}}, E_{\text{concept}})$, which organizes functional semantics into a hierarchical structure. This graph serves as a semantic backbone for aligning raw configuration options with high-level interpretable functional concepts (Example in Table 1).

**Formal Definition.** The Concept Taxonomy Graph is formally defined as a directed acyclic graph:

$$\mathcal{G}_{\text{concept}} = (V_{\text{concept}}, R_{\text{concept}}, T_{\text{concept}}) \tag{7}$$

where:

- $V_{\text{concept}}$: the set of semantic concept nodes, organized into a hierarchical structure from coarse-grained domains to fine-grained functionalities;

- $R_{\text{concept}} = \{\text{SUB\_CATEGORY}\}$: the set of relations that denote parent-child relationships in the hierarchical taxonomy.

- $T_{\text{concept}} \subseteq V_{\text{concept}} \times R_{\text{concept}} \times V_{\text{concept}}$: the set of valid triples representing parent-child entities and relationships in the graph.

This formalization supports multi-level semantic reasoning and enables seamless alignment with configuration items through mapping relations.

**Graph Construction** The Concept Taxonomy Graph is constructed based on a hybrid approach combining expert knowledge and LLM-guided refinement. It consists of two primary components:

- Concept Nodes ($V_{\text{concept}}$): Represent semantic units at varying levels of abstraction, ranging from high-level domains (e.g., Security Features, Cryptography) to fine-grained functionalities (e.g., Process Signal Control).
- SUB_CATEGORY Relations ($R_{\text{concept}}$): Directed edges that denote hierarchical specialization between concepts, enabling multi-level reasoning.

These components are populated through a two-stage process:

**Stage 1: Expert-Defined Coarse-Grained Concepts.**
> We begin with a set of high-level functional domains, defined by kernel experts based on their analysis of the Linux kernel documentation and subsystem organization (e.g., Security Features, Cryptography). These categories form the coarse-grained nodes of the taxonomy and capture key functional dimensions (Table 1).

**Stage 2: LLM-Guided Fine-Grained Expansion.**
> Building upon this expert-defined coarse-grained hierarchy, we guide LLM to incrementally discover and integrate fine-grained concepts. Each new concept is automatically mapped to the taxonomy via LLM-recommended hierarchical paths using SUB_CATEGORY relations, ensuring both semantic coherence and hierarchical consistency.

The Concept Taxonomy Graph is constructed through a hybrid approach combining expert-defined coarse-grained domains with LLM-guided fine-grained expansion. Its hierarchical structure not only captures multi-level functional semantics but also supports dynamic evolution as new functionalities emerge.

This structured yet extensible schema ensures both interpretability and scalability in modeling kernel functionality, laying the groundwork for semantic alignment with configuration items and supporting various downstream reasoning tasks.

## B.2 LLM-KG Synergy Concept Mining

### B.2.1 Mapping Formalization

To bridge the semantic gap between the Configuration Graph and the Concept Taxonomy Graph, we formalize the semantic mapping $\mathcal{M}$ as:

$$\mathcal{M} \subseteq \mathcal{C}_{\text{ext}} \times \mathcal{V}_{\text{cfg}} \times [0, 1] \times \mathcal{R}_{\text{trace}} \times H_{\text{insert}} \tag{8}$$

where:

- $\mathcal{C}_{\text{ext}} = V_{\text{concept}} \cup \{f_{ij}\}$: the extended concept set, combining manually expert-defined coarse-grained concepts and fine-grained sub-concepts $f_{ij}$ generated through LLM-guided mining;

- $\mathcal{V}_{\text{cfg}}$: the set of configuration nodes from the Configuration Graph;

- $[0, 1]$: confidence scores derived from LLM outputs;

- $\mathcal{R}_{\text{trace}}$: a set of reasoning traces or rationales that justify each mapping decision.

- $H_{\text{insert}} = \{h \mid h = (f_{ij}, \text{SUB\_CATEGORY}, c), f_{ij} \in \{f_{ij}\}, c \in \mathcal{C}_{\text{ext}}\}$: a set of hierarchical insertion relations that define how new concepts are integrated into the taxonomy.

Each mapping $(c, v, s, r, h) \in \mathcal{M}$ represents an association between a configuration node $v$ and a concept $c$, assigned a confidence score $s$ and supported by a reasoning trace $r$. The hierarchical relation $h \in H_{\text{insert}}$ further specifies where the new concept should be placed within the Concept Taxonomy Graph (Prompt shown in Figure 3). It typically takes the form of a SUB_CATEGORY link to a concept already present in $\mathcal{C}_{\text{ext}}$, whether that concept was defined by experts or previously mined through LLM inference.

This mechanism enables both semantic alignment with configuration items and structural expansion of the Concept Taxonomy Graph. The mapping is initialized as $\mathcal{M} \leftarrow \emptyset$, with the expert-defined taxonomy preloaded into the LLM prompt to bootstrap the semantic mining process.

### B.2.2 TRAVERSAL STRATEGY

To enable context-aware semantic reasoning over the deep and intricate Configuration Graph (with over 17K nodes spanning 9 nested levels, as detailed in Section 4.2), we propose a two-phase traversal strategy called *Hierarchical Hybrid Traversal (HHT)*. This strategy not only defines an efficient exploration order but also embeds structural priors that guide LLM-based concept mining and mapping. The strategy consists of two complementary phases:

**Global Layer-wise Enumeration.** We first perform a breadth-first search (BFS) from the configuration root (the top-level Linux kernel configuration entry point) to identify all structural branching points (menu, menuconfig, choice), assigning each a depth level $\ell$. This global enumeration establishes a hierarchical encoding that serves as the foundation for localized, structure-aware reasoning in subsequent phases.

**Local Branch-wise Expansion.** For each branch identified in the global phase, we perform a local depth-first search (DFS) to collect its descendant config nodes, forming a *Branch Unit*. Each unit is enriched with two types of contextual information:
- *Parent State ($P_\ell$)*: captures the semantic context inherited from the parent node at level $\ell$, representing high-level configuration intent;
- *Sibling State ($S_\ell$)*: aggregates shared semantics among sibling nodes, providing contrastive cues for fine-grained concept differentiation.

This dual-state representation enables localized, structure-aware LLM reasoning, allowing the model to build upon previously analyzed concepts while maintaining coherence within the taxonomy.

The HHT traversal strategy provides both a structured exploration mechanism and a semantic context framework. By maintaining parent and sibling states throughout the graph, it supports incremental concept mining, where newly discovered concepts are refined based on previously mapped concepts and their hierarchical placements. This design ensures that the knowledge graph evolves coherently, preserving semantic alignment and structural integrity during LLM-driven expansion.

### B.2.3 CONCEPT MINING VIA LLM-KG SYNERGY

We formulate the concept mining process as a node-wise semantic alignment task, guided by the traversal structure introduced in HHT. At each configuration item $v \in \mathcal{V}_{\text{cfg}}$, the LLM generates candidate concept associations by utilizing both the Configuration Graph structure and dynamically updated contextual priors.

Our goal is to construct mappings within the extended concept space $\mathcal{C}_{\text{ext}}$, defined as: $\mathcal{M} \subseteq \mathcal{C}_{\text{ext}} \times \mathcal{V}_{\text{cfg}} \times [0, 1] \times \mathcal{R}_{\text{trace}} \times H_{\text{insert}}$(Def. in Equation 8), where each mapping reflects not only the functional intent of the configuration but also the hierarchical position of concepts within the Concept Taxonomy Graph. To achieve this objective, we further define a structured knowledge context for each configuration node $v$:

$$\mathcal{K}_v = \{\mathcal{P}_v, \mathcal{D}_v, P_\ell, S_\ell\} \tag{9}$$

Where:

- $\mathcal{P}_v$: Hierarchical path from configuration root to node $v$, encoding structural position information;
- $\mathcal{D}_v$: Node-level semantic description (e.g., help text), providing explicit semantic cues;
- $P_\ell$: Parent branch's semantic state derived from LLM analysis;
- $S_\ell$: Aggregated semantics of the sibling nodes in the branch unit, supporting contrastive reasoning.

Given $\mathcal{K}_v$, the LLM predicts a set of candidate concepts:

$$C_v = \{c \in \mathcal{C}_{\text{ext}} \,|\, \text{LLM-conf}(c \,|\, \mathcal{K}_v) \geq \theta\} \tag{10}$$

where $\theta = 0.65$ serves as a confidence threshold to filter low-quality predictions. The resulting mappings associate each configuration item with a set of predicted concepts, along with corresponding confidence scores and rationales explaining the associations. Additionally, the LLM generates recommended hierarchical insertion paths, which are encoded as SUB_CATEGORY relations to facilitate seamless integration into the Concept Taxonomy Graph.

The validated mappings are then incorporated into the global mapping set $\mathcal{M}$ after human-in-the-loop validation (Section 2.3), and corresponding RELATED_TO relations are established between

configuration items $v \in \mathcal{V}_{\text{cfg}}$ and their associated concepts $c \in \mathcal{C}_{\text{ext}}$. This process enriches the heterogeneous graph system with interpretable semantic links.

This LLM-KG synergy concept mining process establishes interpretable mappings between configuration items and functional concepts, guided by structural priors from the Configuration Graph. It not only identifies semantically meaningful concepts but also integrates them into the Concept Taxonomy Graph through hierarchical insertion paths.

### B.3 PROMPT AND REASONING TRACE

To enhance transparency and reproducibility of the *LLM-KG Synergy Concept Mining* process (Section 2.1), we provide representative examples of the prompts given to the LLM and the corresponding reasoning traces it produced.

Figure 3 shows an example prompt designed for extracting functional semantics from a kernel configuration option. The prompt incorporates structured context from the configuration graph and domain-specific metadata, guiding the LLM to infer candidate concepts with confidence scores.

## Prompt

**System Prompt**
You are a Linux kernel configuration expert.
Your goal is to help build a **consistent, human-readable, and hierarchical** concept taxonomy to support understanding and navigation of kernel configs.

**Mining Prompt**
Your task is to propose **0–5 general-purpose concepts** for the following kernel configuration item, based on the following information:
Instructions:
1. **Inheritance and Specialization**:
   – Prioritize inheriting from the parent node's concept(s)(If existing), by reusing or specializing them into a more fine-grained subcategory.
     – Example: If parent concept is `Memory Management`, child config may be labeled `Swap` or `Hugepage`.
     – Ensure the new concept remains within the same conceptual branch.
2. **Sibling Alignment (Optional)**:
   – If sibling configs already have concepts, consider **reusing or generalizing** from them when applicable.
   – Keep concept usage consistent among related configs.
3. **Relevance and Generalization**:
   – Propose concepts that reflect **real functional meaning** (e.g., kernel features, subsystems, hardware usage, performance domains).
   – Avoid overly narrow, hardware-specific, or ad hoc tags (e.g., `x86_special_irq`, `ath10k_debug_flag`).
   – Instead, generalize (e.g., `Interrupt Handling`, `Wireless Debugging`).
4. **Hierarchy Fit**:
   – All suggested concepts must map to the **existing concept hierarchy** (`suggested_concept_path`) provided below.
   – Do **not create new concept categories** beyond the existing hierarchy.
5. **Confidence Score**:
   – Assign a `confidence` value (0.0–1.0) for each concept:
     – `> 0.85`: Strong match (clear functional + hierarchical fit)
     – `0.6–0.85`: Moderate confidence
     – `< 0.6`: Weak — avoid unless no better alternatives
---
Reference concept hierarchy:
{predefined_taxonomy}

Optional parent analysis and sibling status:
{contextual_status}

Configuration Information:
{configuration, help text}
Rules:
Output Format:
Return a list of relevant concepts or `[]` if nothing suitable applies.
Output Format:
[
    {
      "concept_name": "suggested_concept",
      "suggested_concept_path": "parent1 –> parent2 –> suggested_concept",
      "reason": "Explain why this concept fits the configuration",
      "confidence": "Confidence score, following standards"
    },
    .....
]

Figure 3: Prompt for LLM-KG Synergy concept mining.

```
Output Example

"BLK_RQ_ALLOC_TIME":
[
    {
        "label_name": "Storage Support",
        "suggested_label_path": "Hardware Support –> Storage Support",
        "reason": "Inherits from parent node labels which strongly emphasize storage–
related functionality. This config item deals with block request allocation timing, which
is a core storage subsystem mechanism.",
        "confidence": 0.85
    },
    {
        "label_name": "I/O Optimization",
        "suggested_label_path": "Performance –> I/O Optimization",
        "reason": "Alignment with sibling configurations (BLK_ICQ,
BLK_DEV_BSG_COMMON) that share I/O performance characteristics. The allocation
timing optimization fits within the performance domain.",
        "confidence": 0.8
    }
]
]
```

Figure 4: LLM reasoning output for concept mining.

Figure 4 illustrates the reasoning trace generated by the LLM in response to this prompt. The trace includes intermediate steps, justifications, and the final mapping decision, which are then verified and integrated into the evolving concept taxonomy (Appendix B.4).

Together, these examples demonstrate how ICMOS leverages prompt engineering and structured reasoning to produce interpretable and verifiable mappings from configuration items to semantic concepts.

## B.4 Concept Taxonomy

We construct an expert-defined taxonomy of kernel configuration concepts, organized into eight high-level categories with corresponding subcategories (Table 8). This taxonomy serves as the seed structure for OSKC-KG and can be extended through incremental concept mining.

Table 8: Expert-defined Concept Taxonomy

| Category | Subcategories |
| --- | --- |
| Core Subsystem | Hotplug, Schedule, Power Management, Debugging, Virtualization, Containerization, Swap, Filesystem, Networking, Logging & Tracing, Cgroup, Hugepage |
| Kernel Mechanisms | Tracing, Isolation Mechanisms, Namespaces |
| Security Features | Access Control, Integrity & Verification, Sandboxing & Isolation, Cryptography |
| Hardware Support | CPU Support, GPU & Display Support, Storage Support, Network Adapters Support, Peripheral Support, Multimedia Support, Embedded & SoC Support, Power & Thermal Management |
| Performance | CPU Optimization, Memory Optimization, Disk Optimization, Network Optimization, Latency Optimization, Energy Efficiency Optimization, Real-Time Optimization, I/O Optimization |
| Build & Boot | Bootloader Support, Initramfs, Compression |
| Compatibility | Legacy Support, POSIX Compliance |

To illustrate the process of incremental concept mining, we provide an example for the *Hotplug* category under the *Core Subsystem*. Figure 5 shows a partial view of the mined subcategories, which were automatically extracted and refined through ICMOS. This example demonstrates how ICMOS extends the expert-defined taxonomy with finer-grained semantic concepts, such as CPU hotplug, memory hotplug, and Device hotplug. The complete multi-level taxonomy, including all mined concepts, will be released as part of our open-source OSKC-KG to facilitate reproducibility and future research.

Figure 5: Example of mined concepts

## B.5 OSKC-KG STATISTICS OVERVIEW

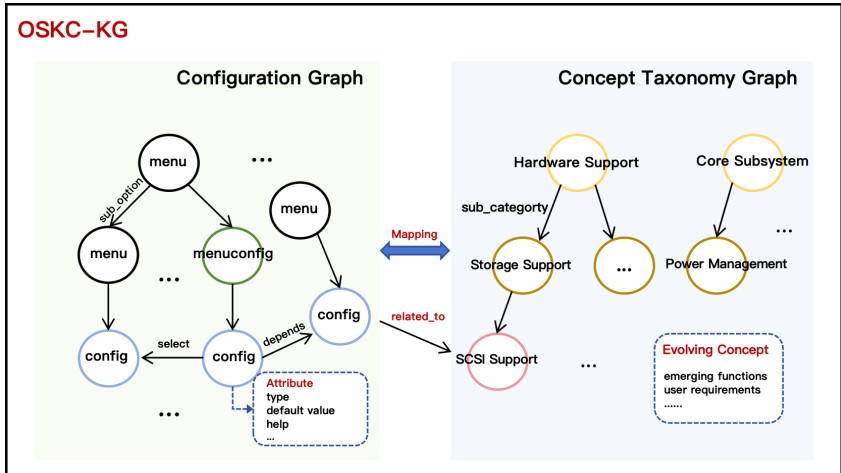

Figure 6: OSKC-KG: Node-Relation Graph Schema

Based on our ICMOS framework, we build a comprehensive knowledge graph OSKC-KG that captures the rich semantics of kernel configurations while supporting extensibility and interoperability. *Figure 6* illustrates the node-relation schema of OSKC-KG.

**Implementation Details**    We implement the ICMOS framework using *DeepSeek-V3-0324* as the core LLM, selected for its strong performance in technical domain reasoning and multilingual support. We use a custom prompting strategy to extract semantic mappings (see Figure 3 for prompt templates). The resulting knowledge graph is stored in *Neo4j*, leveraging its native graph storage and query capabilities for efficient traversal, semantic retrieval, and dynamic expansion.

**OSKC-KG Statistics**    To ensure broad applicability and reflect both stability and evolution trends, we construct the OSKC-KG by analyzing the evolving Linux 6.x kernel series (versions 6.1–6.15), covering 17,624 representative configuration options from a broader pool of 16,725–18,315 version-specific variants. This range captures both stability and evolution trends across recent kernels, ensuring broad applicability.

By integrating these with a taxonomy of 1,303 fine-grained semantic concepts via LLM-KG synergy mining, we obtain a graph comprising 18,927 nodes and 145,191 edges. Notably, 96.5% of configuration items are annotated with hierarchical semantic concepts, and each item is associated with an average of 3.9 concepts, including those inherited from parent nodes across multiple levels. These results highlight the scalability and expressiveness of OSKC-KG, laying a strong foundation for downstream reasoning, retrieval, and optimization tasks. The complete statistics are reported in Table 2.

Furthermore, our framework supports dynamic evolution: new configuration items can be seamlessly integrated into the graph under appropriate menu paths, while emerging functional semantics can be incrementally added to the concept taxonomy. These characteristics, namely extensive coverage, semantic richness, and structural extensibility, establish OSKC-KG as a robust foundation for intelligent reasoning over complex kernel configuration spaces.

## C  CONFIGURATION QA DATASET

To enable reproducible evaluation of semantic retrieval performance, we construct a human-annotated ground truth dataset comprising 100 QA pairs, curated by Linux kernel domain experts. The dataset is designed to evaluate two core capabilities:

1. **Concept Understanding**: Given a configuration item (e.g., `CONFIG_BFQ_GROUP_IOSCHED`), predict its associated high-level semantic concepts (e.g., "I/O Optimization", "Scheduler").

2. **Configuration Selection**: Given a functional intent (e.g., "Improve file I/O performance"), select the most relevant configuration options from a provided candidate list.

To facilitate future research and ensure full reproducibility, we plan to open-source this QA dataset, along with the complete OSKC-KG and our evaluation scripts, upon paper acceptance.

Illustrative examples for both QA types are provided in Figure 7. We use this dataset to quantitatively compare the performance of the LLM-only baseline and the OSKC-KG augmented pipeline, as reported in Section 4.3.

Type: Concept Understanding

Question : "According to the {concept taxonomy}, which functional concepts does the CONFIG_CFS_BANDWIDTH configuration option involve?"

Answer   : ["Resource Control", "Fair Scheduler"]

Type: Configuration Selection

Question : "Which kernel configuration options directly relate to containerization technologies and namespace isolation mechanisms? "

Options   : ["CONFIG_TIME_NS", "CONFIG_KCMP", "CONFIG_PID_NS", "CONFIG_UTS_NS"]

Answer   : ["CONFIG_TIME_NS", "CONFIG_PID_NS", "CONFIG_UTS_NS"]

Figure 7: QA Dataset Example

## D  CONFLICT-AWARE VALIDATION MECHANISM

To enhance the reliability of intelligent configuration optimization guided by the LLM agent, we introduce a *Conflict-Aware Validation Mechanism* that augments the reasoning layer with additional symbolic constraints from OSKC-KG.

Beyond semantic relevance, the LLM agent is guided to infer about feasibility by incorporating configuration-level structural and behavioral priors, including:

- *Dependency and Selection Constraints*: (depends on, selects) to ensure compatibility across configuration items.
- *Default Values and Value Types*: to support informed value prediction and avoid invalid assignments.
- *Visibility and Activation Conditions*: to filter out invisible or context-clashing configurations.

These constraints are encoded into the prompt during reasoning. Additionally, we require the LLM agent to filter out configuration items that could impact operating system boot or compilation.

This mechanism avoids generating infeasible or conflicting configuration suggestions, improves assignment recommendations for optimization objectives (e.g., File I/O, MySQL Latency), and ensures that all recommended configurations are semantically meaningful and practically deployable.

# E  AGENTIC REASONING PROMPTS AND OPTIMIZATION OUTPUTS

To complement Section 2.2 and Section 4.4, we provide illustrative examples of the prompts used for concept-oriented agentic reasoning and the resulting configuration optimization outputs.

Figure 8 shows a representative prompt template used to guide the LLM agent in generating optimization suggestions for candidate configuration parameters. The prompt incorporates structured context from OSKC-KG to constrain the LLM's output and reduce hallucinations.



**Performance Optimization prompt**

**As a Linux kernel performance optimization expert.**

Your task is to analyze a given kernel configuration item and determine whether it should be adjusted to better support
**{target}** performance optimization**
Given configuration item comes with its metadata (value type, default value, help text, depends on info, select configs),

**Follow the steps below carefully:**

1. **Understand the Configuration**:
    – Identify the configuration **value type** and its **default value**.
    – Analyze the core functionality of each configuration item based on knowledge of kernel subsystems and config help information.
    – Consider the impact of potential dependency and selection configuration items.
2. **Optimization Decision**:
    – Check if the **default value** is already optimal for `{target}`.
    – If it's **suboptimal**, suggest a **new value** and explain the reason **internally (not in the output)**.
    – If the config is **essential for boot or system stability**, skip it by returning `{}`.
–––
Exclusion Criteria:
Return `{}` if **any** of the following apply:
– It is **unrelated** to `{target}` performance.
– It is **critical for system boot, init, or core stability**.
– It must remain at its default to **avoid build failures**.

Output Format:
Only output a JSON object, where the key is the config name, and the value is the new recommended setting.

If optimized:
Example Output:
{
    "CONFIG_NAME": "recommended_value"
}

Only return the JSON object. No additional explanations.
Below are the configuration information:
{config_info}



Figure 8: File I/O Case Study

Figure 9 illustrates the configuration recommendations generated by ICMOS for optimizing the *file I/O* benchmark. It displays the relevant kernel configurations and their recommended optimized values, which are subsequently validated through expert inspection and benchmarking.

These examples serve two purposes: (i) to demonstrate the interpretability and traceability of the agentic reasoning process, and (ii) to illustrate how optimization recommendations are generated and linked back to semantic concepts in OSKC-KG. A more detailed discussion of these results and their implications is provided in the Case Study (Appendix F).

```
"CONFIG_CACHEFILES": "y",
"CONFIG_CACHEFILES_ONDEMAND": "y",
"CONFIG_FSCACHE": "y",
"CONFIG_FSCACHE_STATS": "n",
"CONFIG_NETFS_SUPPORT": "y",
"CONFIG_JFFS2_FS": "y",
"CONFIG_JFFS2_FS_WRITEBUFFER": "y",
"CONFIG_UBIFS_FS_LZO": "y",
"CONFIG_UBIFS_FS_ZSTD": "y",
"CONFIG_NFS_FSCACHE": "y",
"CONFIG_SQUASHFS_CHOICE_DECOMP_BY_MOUNT": "y",
"CONFIG_SQUASHFS_ZSTD": "y",
"CONFIG_SCSI_SYM53C8XX_DEFAULT_TAGS": "16",
"CONFIG_SCSI_SYM53C8XX_MMIO": "y",
"CONFIG_LZ4_COMPRESS": "y",
"CONFIG_LZ4_DECOMPRESS": "y",
"CONFIG_ZSWAP_COMPRESSOR_DEFAULT_LZ4": "y",
"CONFIG_ZSWAP_ZPOOL_DEFAULT_ZSMALLOC": "y",
"CONFIG_CIFS_SMB_DIRECT": "y",
"CONFIG_CIFS": "y",
"CONFIG_RDMA": "y",
"CONFIG_F2FS_FS": "m",
"CONFIG_NO_HZ_FULL": "y",
"CONFIG_HZ_1000": "y",
"CONFIG_BFQ_GROUP_IOSCHED": "y",
"CONFIG_BLK_CGROUP": "y",
"CONFIG_BLK_CGROUP_IOCOST": "y",
"CONFIG_BLK_CGROUP_IOLATENCY": "y",
"CONFIG_BLK_CGROUP_IOPRIO": "y",
"CONFIG_BLK_DEV_THROTTLING": "y"
```

Figure 9: File I/O Case Study

# F    CASE STUDY

Figure 10: File I/O Case Study

To further illustrate the effectiveness and interpretability of our framework, we present a case study focusing on kernel *File I/O* performance optimization. Given the optimization goal of improving I/O efficiency, ICMOS invokes the LLM Agent to interpret the intent and perform structured reasoning over the Concept Taxonomy Graph. Starting from the root node, the agent progressively identifies semantically relevant concepts (e.g., I/O Optimization, Swap). Taking I/O Optimization as an example, the agent retrieves the corresponding configuration subgraph via the RELATED_TO mappings (See the Figure 10). Within this subgraph, the LLM agent performs knowledge-augmented reasoning over the configuration space to generate interpretable and effective configuration optimization suggestions.

Through this reasoning process, the LLM agent recommends two key configuration options: CONFIG _BFQ_GROUP_IOSCHED=y, CONFIG_BLK_DEV_THROTTLING=y, which activate the BFQ I/O scheduler with group-aware control and enable I/O bandwidth throttling, respectively. These settings improve I/O responsiveness under concurrent workloads and help stabilize throughput across multiple tasks. In contrast, AutoOS fails to surface these options due to its lack of structured concept reasoning. It either omits I/O-related tunings or recommends unrelated parameters, leading to suboptimal performance improvements.

This case demonstrates the strength of ICMOS in leveraging semantic knowledge to perform goal-directed configuration traversal and generate more relevant, high-impact recommendations. Notably, the decision process is fully traceable, each configuration is backed by a concept linkage and a reasoning path, enhancing both transparency and trust in system tuning.

