# OpenReview forum: "ICMOS: Incremental Concept Mining for OS Kernel Configuration via LLMs Agentic Reasoning"
_ICLR.cc/2026/Conference — ICLR 2026 Conference Withdrawn Submission_

### Official Review · Reviewer_aZqd · 2025-10-26

**Soundness:** 3
**Presentation:** 3
**Contribution:** 3
**Rating:** 6
**Confidence:** 3

**Summary:**

The paper proposes ICMOS, a framework for Linux kernel configuration that couples an LLM with a domain knowledge graph (OSKC-KG) which combines a parsed configuration graph and a concept taxonomy. ICMOS consists of two phases: LLM-KG synergy mining that traverses the config graph with structured context to map options to concepts; and concept-oriented agentic reasoning to incrementally add new concepts and mappings. The framework is evaluated on configuration retrieval QA and intelligent configuration optimization across system-level and application workloads, and has shown consistent improvement over the LLM-only baseline.

**Strengths:**

-paper is well-written and easy to follow

-nice application of leveraging knowledge graph to reduce hallucination and improve performance. The split between Configuration Graph and Concept Taxonomy, plus explicit mapping edges is neat.

-evaluations across QA, microbenchmarks, and real applications show consistent improvements of the proposed method

**Weaknesses:**

-methodology contribution is relatively on the incremental side (i.e., how to leverage knowledge graph to help reduce LLM hallucination on a specific task)

-variance across seeds/runs and significance tests are absent

**Questions:**

-How ICMOS perform on kernel-intensive domains (e.g., eBPF pipelines, GPUs)?

---

### Official Review · Reviewer_AHcG · 2025-10-26

**Soundness:** 2
**Presentation:** 3
**Contribution:** 1
**Rating:** 2
**Confidence:** 4

**Summary:**

This paper introduces ICMOS, a framework that integrates LLM with a heterogeneous knowledge graph (OSKC-KG) for operating system kernel configuration. The approach is designed to address semantic ambiguities in configuration options through incremental concept mining, which involves a two-phase process of LLM-KG synergy mining and agentic concept evolution, supplemented by human-in-the-loop verification. Empirical evaluation indicates improvements in optimization time, configuration success rate, and tail latency compared to LLM-only baselines.

**Strengths:**

1. Identified semantic comprehension deficiencies in kernel option configuration

2. Designed a dual-layer knowledge graph and conducted semantic Q&A experiments along with kernel configuration tests.

**Weaknesses:**

1. Lack of Novelty: The core ideas and implementation presented in this paper bear a strong resemblance to the prior work "BYOS: Knowledge-driven Large Language Models Bring Your Own Operating System More Excellent" (arXiv, May 2025). However, the authors fail to cite or provide a comparative discussion of this highly relevant work.

2. Limited Experimental Scope: The evaluation is confined to a single virtual machine configuration and one operating system (Ubuntu), failing to cover diverse hardware or alternative systems (e.g., Fedora). Furthermore, the study relies exclusively on the DeepSeek-V3 model, omitting any performance benchmarks against lighter-weight models such as GPT-4o-mini or GPT-3.5.

3. Compromised Automation: The proposed method introduces a manual validation step for the knowledge graph. This "human-in-the-loop" requirement undermines the goal of a fully automated tuning process and compares unfavorably to the end-to-end automation of the AutoOS baseline.

4. Robustness and Scalability Concerns: The framework's reliance on a pre-constructed knowledge graph introduces significant robustness and scalability challenges. The approach's effectiveness is questionable when encountering new, out-of-graph configuration options or when the KG's quality is compromised by LLM hallucinations during its construction.

5. Missing Experimental Details: The paper fails to specify the number of search iterations or a comparable convergence criterion for both the proposed method and the baselines. This omission makes it difficult to assess the experimental rigor and fairly evaluate the efficiency of the different approaches.

**Questions:**

1. Can you provide detailed information on the search time overhead and token costs when optimizing kernel configuration?

2. How should we address the issues of LLM hallucinations and the poor quality of knowledge graphs caused by different contributors?

3. See more details in Weaknesses.

---

### Official Review · Reviewer_RBsy · 2025-10-31

**Soundness:** 2
**Presentation:** 2
**Contribution:** 2
**Rating:** 4
**Confidence:** 4

**Summary:**

This paper proposes ICMOS, a novel framework that integrates large language models (LLMs) with a heterogeneous knowledge graph (OSKC-KG) to perform incremental concept mining and reasoning for Linux kernel configuration. The system aims to address the challenges of the large configuration space (17K+ options), semantic ambiguity, and dependency complexity in kernel tuning. The proposed approach combines three main components: (1) OSKC-KG construction merging configuration and semantic concept graphs, (2) an LLM-KG synergy mining process to align configuration options with functional semantics, and (3) an agentic reasoning mechanism for incremental concept evolution.
The authors conduct extensive experiments on two kinds of tasks — configuration retrieval QA and intelligent configuration optimization — demonstrating significant improvements over LLM-only baselines in both semantic accuracy and system performance. ICMOS claims to halve optimization time and improve configuration success rates substantially.

**Strengths:**

1. Novel integration of LLMs and structured knowledge graphs. The paper successfully combines symbolic reasoning (via OSKC-KG) with LLM-based semantic inference, offering a structured and interpretable approach to kernel configuration understanding.
2. Comprehensive evaluation. The experiments cover both retrieval and optimization tasks, reporting clear quantitative metrics such as UnixBench, latency, and throughput across multiple workloads (Apache, MySQL, Redis, etc.).
3. Strong empirical gains. The reported improvements (e.g., 50% reduction in optimization time — 39 minutes vs. 82 minutes per cycle — and an increase in configuration success rate from 29% to 76%) are notable and suggest substantial practical benefits.
4. Clarity and reproducibility. The methodology is well-documented, with detailed appendices describing the graph construction, traversal strategy, and agentic reasoning process. The inclusion of a reproducibility statement with open-sourcing plans is commendable.
5. Interpretable reasoning pipeline. The human-in-the-loop verification and reasoning trace examples (Appendix B.3–B.4) enhance transparency and make the system more trustworthy for critical system-level use cases.

**Weaknesses:**

1. Limited novelty in the concept-mining methodology. While the integration design is elegant, the idea of combining LLMs with knowledge graphs for structured reasoning has been explored in prior works (e.g., GraphRAG,). The main innovation of this paper lies in its application domain (kernel configuration) rather than a fundamental algorithmic breakthrough.
2. Insufficient details on the conflict-aware validation mechanism. In Table 6, the improvements in optimization time, average conflict count, and configuration success rate appear to stem primarily from the proposed conflict-aware validation mechanism. Although Appendix D briefly describes its implementation, the explanation lacks sufficient technical depth to understand how the mechanism operates and interacts with the reasoning process.
3. Unclear relationship between semantic relevance and performance tuning. The paper demonstrates that improving semantic grounding through ICMOS leads to better optimization outcomes, but it does not clearly explain the causal relationship between semantic relevance and system-level performance gains. A more explicit analysis or ablation would strengthen this connection.
4. Dependence on human verification and lack of hallucination mitigation strategy. The concept taxonomy relies on expert-defined coarse-grained domains and LLM-guided fine-grained concept expansion. However, the paper does not explain how potential LLM hallucinations or misclassifications are detected and corrected during this process. Although expert validation is mentioned, the extent of manual involvement, verification cost, and possible automation strategies are not quantified or discussed, which may limit the scalability and autonomy of the proposed framework.

**Questions:**

1. How sensitive are the performance improvements to the choice of LLM (e.g., DeepSeek-V3 vs GPT-4)? Could smaller models achieve similar results when combined with OSKC-KG?
2. In the optimization experiments, how is the balance maintained between semantic relevance and performance-driven tuning? Does the model ever trade off correctness for speed?
3. How frequently does OSKC-KG need to be updated with new kernel versions? When major kernel updates occur (e.g., significant changes in configuration options or dependency structures), does the framework require a complete reconstruction of the knowledge graph? If so, the associated computational and manual overhead may no longer be negligible, contrary to the current claim of incremental update efficiency.

---

### Official Review · Reviewer_xC6Z · 2025-11-01

**Soundness:** 1
**Presentation:** 2
**Contribution:** 2
**Rating:** 2
**Confidence:** 4

**Summary:**

This paper introduces ICMOS, a framework that combines Large Language Models (LLMs) with a specialized knowledge graph (OSKC-KG) to automate Linux kernel configuration. ICMOS aligns configuration items with their functional meanings through structured graphs and adapts to new concepts with human oversight. Evaluation shows ICMOS outperforms LLM-only approaches, delivering significant improvements in accuracy, system performance, and optimization efficiency.

**Strengths:**

1. Systematic Benchmark for OSKC Tasks: Creation of a dedicated benchmark for evaluating OSKC understanding.
2. Incremental Interpretable Modeling: Achieved through dual-phase LLM-KG collaborative mining and agentic concept evolution.

**Weaknesses:**

1. The evaluation is confined to a virtualized environment, which does not reflect the complexities of modern hardware architectures. This limitation undermines the framework's demonstrated capability for large-scale, high-concurrency scenarios.
2. The evaluation uses only one LLM backbone, failing to demonstrate that the benefits are inherent to the KnowOS framework and not specific to a particular model.
3. The paper does not detail the specific mechanisms used to prevent the LLM hallucinations during the KG's construction and incremental update phases. A quantitative analysis of the noise or error rate in the mined concepts is critically needed.
4. The paper fails to address ICMOS's capability to maintain performance across different kernel versions.
5. High computational costs from using DeepSeek-V3 are ignored, with no inference latency or resource metrics provided.

**Questions:**

1. Can you provide experiments encompassing diverse scenarios?

2. What is the novelty of the knowledge question-answering task compared to previous work?

---

### Note · Authors · 2025-12-12

I have read and agree with the venue's withdrawal policy on behalf of myself and my co-authors.